# A Clinical Insight on New Discovered Molecules and Repurposed Drugs for the Treatment of COVID-19

**DOI:** 10.3390/vaccines11020332

**Published:** 2023-02-01

**Authors:** Surojit Banerjee, Debadri Banerjee, Anupama Singh, Sumit Kumar, Deep Pooja, Veerma Ram, Hitesh Kulhari, Vikas Anand Saharan

**Affiliations:** 1School of Pharmaceutical Sciences and Technology, Sardar Bhagwan Singh University, Balawala, Dehradun 248001, India; 2Department of Pharmaceutical Sciences, Central University of Haryana, Jant-Pali, Mahendergarh 123031, India; 3School of Pharmacy, National Forensic Sciences University, Sector 9, Gandhinagar 382007, India; 4School of Nano Sciences, Central University of Gujarat, Sector 30, Gandhinagar 382030, India

**Keywords:** SARS-CoV-2, antiviral, monoclonal antibody, cytokine blockers, correlates of protection

## Abstract

Severe acute respiratory syndrome coronavirus 2 (SARS-CoV-2) began churning out incredulous terror in December 2019. Within several months from its first detection in Wuhan, SARS-CoV-2 spread to the rest of the world through droplet infection, making it a pandemic situation and a healthcare emergency across the globe. The available treatment of COVID-19 was only symptomatic as the disease was new and no approved drug or vaccine was available. Another challenge with COVID-19 was the continuous mutation of the SARS-CoV-2 virus. Some repurposed drugs, such as hydroxychloroquine, chloroquine, and remdesivir, received emergency use authorization in various countries, but their clinical use is compromised with either severe and fatal adverse effects or nonavailability of sufficient clinical data. Molnupiravir was the first molecule approved for the treatment of COVID-19, followed by Paxlovid™, monoclonal antibodies (MAbs), and others. New molecules have variable therapeutic efficacy against different variants or strains of SARS-CoV-2, which require further investigations. The aim of this review is to provide in-depth information on new molecules and repurposed drugs with emphasis on their general description, mechanism of action (MOA), correlates of protection, dose and dosage form, route of administration, clinical trials, regulatory approval, and marketing authorizations.

## 1. Introduction

Novel coronavirus disease 2019 (COVID-19), also known as severe acute respiratory syndrome coronavirus 2 (SARS-CoV-2), has become a grave issue in today’s world [1]. Initially, SARS-CoV-2 infection was limited to the Republic of China, but with time, its infection spread across the entire globe. Although the peak of COVID-19 is on the decline after a long time, still cases of infection are increasing due to the emergence of new strains, resulting in changes in symptoms and treatment strategies [2]. A grave challenge with COVID-19 is the continuous mutation of SARS-CoV-2 virus leading to several variants (B.1.1.7, B.1.351, P.1, B.1.617.2, B.1.1.529 etc.) spread all over the world [3]. To control the situation, drug regulatory agencies (such as the United States Food and drug Administration (USFDA) and the European Medicines Agency (EMA)) have given emergency use approval (EUA) to several novel molecules and pre-existing drugs for controlling the pandemic. Research on COVID-19 vaccines and drugs is still ongoing to find the best cure. Different vaccines, such as ChAdOx1, BNT162 (especially BNT162b2), Covaxin, mRNA-1273, and AD26.COV2.S, have been found effective against different strains of SARS-CoV-2 [3]. However, the effect of vaccines on boosting immunity varies for different strains and their variant. Research funding agencies are aggressively collaborating with pharmaceutical companies and research institutions for the development of vaccines and drugs for the treatment of COVID-19.

Vis-a-vis a vaccine development program, pharmaceutical companies are also aggressively working to discover new antivirus molecules against COVID-19. The quest for finding a treatment started with repurposing efforts on approved drugs. Some pre-existing drugs have shown good therapeutic efficacy against SARS-CoV-2 [1]. Repurposing of drugs is advantageous due to the reduced cost and speeding up of the approval process from regulatory authorities. However, several failures have been observed as in the case of remdesivir, chloroquine, and hydroxychloroquine [4,5]. These drugs either showed adverse effects at higher doses or were effective only in combination with other drugs [6,7,8,9]. EMA has authorized some of these drugs for their emergency use and for the conduct of clinical trials [10,11,12].

However, like developed vaccines, new molecules and repurposed drugs work by boosting the human immune system [2]. These new molecules and repurposed drugs are directly or indirectly associated with the inhibition of the mitogen-activated CD3^+^, CD4^+^, and CD8^+^ T cell proliferation. Regulatory T cells are also controlled by these repurposed drugs and new molecules. In the absence of other immune cells, Wharton’s jelly mesenchymal stem cells inhibit the proliferation of alloreactive CD4^+^ and CD8^+^ T cells. Wharton’s jelly mesenchymal stem cells also inhibit the proliferation of B cells and T-cell-secreted interferon-γ. Different monoclonal antibodies (MAbs) (such as sarilumab, tocilizumab, and baricitinib), repurposed drugs (such as dexamethasone, hydroxychloroquine, and naproxen), and convalescent plasma inhibit IL-6, IL-10 (responsible for T-cell and B-cell stimulation) and serve as anti-inflammatory agents and anticytokines. They are involved in diminishing C-reactive protein and ferritin, inflammatory markers responsible for COVID-19 disease. This review critically investigates various pharmaceutical formulations, their mechanism of action, correlates of protection, dose and dosage form, route of administration, clinical trials, regulatory approval, and marketing authorizations of repurposed drugs and new molecules.

## 2. New Discovered Molecules

New molecules/drugs are discovered by humans for the therapeutic management of new diseases or to address existing health challenges. New molecules/drugs require extensive clinical investigations, followed by approval from drug regulatory agencies for marketing. Several new molecules have been discovered to combat SARS-CoV-2. The newly discovered molecules have been tested in silico via molecular docking studies to assess their interaction with the target receptor on SARS-CoV-2. These new molecules are either antivirals or immunomodulators [13,14]. It has been observed in different studies that these molecules can block SARS-CoV-2 in their conventional way. For example, molnupiravir blocks RNA-dependent RNA polymerase (RdRp) to stop the replication of SARS-CoV-2. Novel MAbs (both alone and in combination) have also been investigated in clinical trials. Drug regulatory authorities have approved several new drugs via the EUA route. Some of these new drugs or molecules are now available commercially for the treatment of COVID-19.

On the other hand, research on new drugs, both monotherapy and combination therapy, is also ongoing. In 2021, the World Health Organization (WHO) recommended EUA to two new drugs, baricitinib and sotrovimab, and later, full approval was granted in 2022 [10].

### 2.1. Molnupiravir

#### 2.1.1. General Description

Molnupiravir (EIDD-2801/MK-4482) is a ribonucleoside prodrug of N-hydroxycytidine [11,12,13]. It was developed by Drug Innovation Ventures at Emory University and later procured by Ridgeback Biotherapeutics in collaboration with Merck, US. It is a prescription antiviral drug and is available as pills, inhaled powder, liquid, and intravenous solutions.

#### 2.1.2. Mechanism of Action

This novel antiviral agent blocks RdRp of the SARS-CoV-2 virus to inhibit the viral genome transcription and replication [14,15,16,17,18]. It was reported that molnupiravir becomes cleaved in plasma and forms β-D-N4-hydroxycytidine (NHC). NHC is then distributed and metabolized into NHC triphosphate in the cytoplasm of the liver cells, which acts as the substrate of RdRp. NHC causes mutation in the viral RNA with the help of RdRp and changes the genome sequence, leading to the inhibition of genome transcription and replication for SARS-CoV-2.

#### 2.1.3. Clinical Trial, Route of Administration, Dose, and Dosage Form

In preclinical trials, molnupiravir showed effect on the blocking replication of SARS-CoV-2 in mice models [19]. It was found safe and efficacious in ferrets at 2.3 and 7 mg/kg of body weight when administered BID orally [16]. In a different study, the administration of molnupiravir in a Syrian hamster showed blocking of the viral replication of SARS-CoV-2 [20].

The phase 1 randomized, double-blind, and placebo controlled trial (NCT04392219), in 130 adult human volunteers divided into four groups, for molnupiravir was conducted by the developer, Ridgeback Biotherapeutics, at the Covance Leeds Clinical Research Unit, UK [14,21,22,23]. The first group received a single oral-dose capsules of 50 to 1600 mg in fasted state, the second group was administered two oral-dose capsules of 200 mg in fed or fasted state, and the third group received twice daily an oral-dose capsule of the drug or placebo. It was observed that molnupiravir was more tolerable than placebo, and adverse events were also lesser than placebo. Phase 2 (NCT04405570) randomized, double-blind, and placebo-controlled clinical trial, in 204 adults and older adults divided into four groups, was conducted by Ridgeback and Merck at different locations in the US [14,21,22,24]. The groups were administered with a 200, 400, or 800 mg BID oral capsule for 5 days. Placebo was taken as oral capsule BID for 5 days. Viral replication was observed in 1.9% participants treated with molnupiravir and 16.5% participants who received placebo. Moreover, 400 and 800 mg BID doses were found effective in reducing viral replication with no significant adverse effects. In another phase 2/3 randomized, double-blind, and placebo-controlled trial (NCT04575597), in 1850 adults and older adults at 200, 400, or 800 mg BID oral dose and placebo every 12 h for 5 days, was conducted by Merck at 173 different locations in the US, Brazil, Chile, Argentina, Canada, Colombia, Egypt, France, Germany, Mexico, and so on [11,12,25]. It was observed that molnupiravir was effective in reducing death, 6.8% (95% CI) for the drug and 9.7% (95% CI) for the placebo, in hospitalized patients. Adverse events were alleviated with the drug (30.4%) when compared with the placebo (33%). Unfortunately, this trial was terminated by the data safety monitoring board (DSMB) probably because of the doubtful beneficial data of the drug as compared with the placebo.

#### 2.1.4. Regulatory Approval and Marketing Authorization

Molnupiravir was approved by the United Kingdom Medicines and Healthcare Products Regulatory Agency (MHRA) in November 2021. MSD (Merck Sharp & Dohme) markets this drug as 200 mg hard capsules in the UK as Lagevrio^®^ [26,27]. EMA (November 2021) gave EUA to it and recommends its administration to patients who do not need supplemental oxygen [28]. EUA was also granted by USFDA (December 2021) for molnupiravir [29]. Sun Pharma, Cipla and Torrent market it as Molxvir^®^, Cipmolnu^®^, and Molnutor^®^, respectively, in India after obtaining DCGI (Drug Controller General of India) approval [30,31,32].

### 2.2. Paxlovid™

#### 2.2.1. General Description

Paxlovid™ is a combination of two drugs, nirmatrelvir and ritonavir, developed by Pfizer [33]. It is a protease inhibitor, which is intended for oral administration.

#### 2.2.2. Mechanism of Action

Nirmatrelvir blocks the SARS-CoV-23-CL protease in the proteolysis phase, leading to the inhibition of the replication of the virus. It is administered with a lower dose of ritonavir, which is a strong CYP3A4 inhibitor and HIV protease inhibitor [34]. Therefore, this combination results in the synergistic inhibition of SARS-CoV-2 viral protease [33].

#### 2.2.3. Clinical Trial, Route of Administration, Dose, and Dosage Form

The phase 1 open-label, randomized, single-dose, and crossover clinical trial (NCT05263921) for Paxlovid™ was conducted in 12 adults (divided in different groups) [35]. In some groups, participants were administered oral commercial tablets in fasted state, whereas in other groups, participants received oral powder. Nirmatrelvir (150 mg)/ritonavir (100 mg) in combination was orally administered in fasted state with water, with applesauce, or with vanilla pudding to the participants. An EPIC-HR (evaluation of protease inhibition for COVID-19 in high-risk patients) study, EPIC-SR (evaluation of protease inhibition for COVID-19 in standard-risk patients) study, and EPIC-PEP (evaluation of protease inhibition for COVID-19 in postexposure prophylaxis) studies have been conducted to assess the clinical benefits of Paxlovid™ [33]. The EPIC-HR study (NCT04960202) was a phase 2/3 randomized and quadruple-blinded trial, conducted across Europe, Africa, North and South America, and Asia in 2246 adults and older adults, was divided into different studies; that is, in that trial, the participants were given the drug combination and placebo through the oral route as a tablet and/or capsule. According to the interim analysis result of Pfizer, the drug combination showed about 89% risk of hospitalization, and chances of death were decreased when compared with the placebo group. Adverse events were less with the drug combination rather than in the placebo, that is, 1.7% and 6.6%, respectively. Final analysis showed that the therapeutic efficacy was maintained in 1379 patients with a percentage point difference of −5.81 (95% CI, −7.78 to −3.84; *p* < 0.001) [36].

#### 2.2.4. Regulatory Approval and Marketing Authorization

Paxlovid™ received its EUA from EMA in December 2021 [37], and EUA from USFDA was obtained in December 2021 [33]. In India, Torrent Pharma and Aurobindo Pharma have been licensed to commercialize this product [38].

### 2.3. Baricitinib

#### 2.3.1. General Description

Baricitinib is an orally active immunomodulator, anti-inflammatory, and anticancer agent [39,40].

#### 2.3.2. Mechanism of Action

Baricitinib selectively blocks JAK 1/2 and activates the JAK-STAT (signal transducers and activators of transcription) signaling pathway. It also inhibits numb-associated kinase (NAK) family enzymes, which include BMP-2-inducible kinase (BIKE), serine/threonine-protein kinase 16 (STK16), adaptor-associated kinase 1 (AAK1), and cyclin G-associated kinase (GAK). This results in the blocking of the AP-2 scaffolding protein, which is responsible for the entry and propagation of SARS-CoV-2 [41]. On the other hand, baricitinib also acts as anticytokines and reduces the inflammatory markers of COVID-19 disease, which includes C-reactive protein (CRP), IL-6, and ferritin.

#### 2.3.3. Clinical Trial, Route of Administration, Dose, and Dosage Form

Eli Lilly conducted and completed the phase 3 (NCT04421027) trial, a randomized, double-blinded, and parallel-assigned study (COV-BARRIER), on baricitinib in 1525 adults and older adults at 96 different locations in the US, Argentina, Republic of Korea, Brazil, Mexico, Germany, India, Japan, Italy, Russian Federation, Puerto Rico, Spain, and UK to estimate the therapeutic efficacy of baricitinib in hospitalized COVID-19 patients [42]. Participants were administered 4 mg baricitinib once daily through the oral route against the standard of care. As a result, the chance of mortality was found to be reduced by 38.2%, and one death was found averted per 20 participants. It was also observed that in the baricitinib-treated group, the mortality was 10% in 60 days (*n* = 79; 95% CI; *p* = 0.0050), and in the placebo group, it was 15% (*n* = 116; 95% CI; *p* = 0.0050). Serious adverse events were seen to be slightly lower in the case of the baricitinib-treated group than in the placebo group, that is, 15% and 18%, respectively.

#### 2.3.4. Regulatory Approval and Marketing Authorization

Recently, Eli Lilly received EUA from USFDA (October 2022) for their product Olumiant^®^ (Baricitinib) [43]. EMA has started (in 2021) and is continuing the evaluation of Olumiant^®^ (baricitinib) in the treatment of COVID-19 [44].

### 2.4. Wharton’s Jelly Mesenchymal Stem Cells (WJ-MSC)

#### 2.4.1. General Description

WJ is a gelatinous tissue found in the umbilical cord. It contains myofibroblast-like stromal cells [45]. MSCs are stromal cells, having the capability of self-renewal and multilineage proliferation [46]. MSCs are isolated from the umbilical cord, bone marrow, endometrial polyps, and so on. However, WJ is considered to be the pivotal source of MSCs [47]. MSCs are also considered medicinal signaling cells, which are therapeutically effective stomatal cells. They can be classified in different types of cells, including myocytes, osteoblasts, chondrocytes, and adipocytes. MSCs are found in the cord cells, amniotic fluid adipose tissue, bone marrow, and molar cells. It had proved its effectiveness previously for the treatment of autoimmune diseases, such as Crohn’s disease, systemic lupus erythematosus (SLE), multiple sclerosis, osteoarthritis, and graft versus host disease [48]. Apart from this, it also has immunomodulatory effects and antimicrobial effects. WJ-MSCs are obtained from the cord tissue of newborns and cultured to intensity for MSCs. Then these are placed in saline solution (25 mL) containing 0.5% human serum albumin [49,50].

#### 2.4.2. Mechanism of Action

WJ-MSC curbs mitogen-induced T-cell responses rather than other MSCs. Along with this, WJ-MSC blocks the multiplication of mitogen-activated CD3^+^, CD4^+^, and CD8^+^ T cells [51,52,53]. Apart from peripheral blood mononuclear cells (PBMC) and interferon-γ (IFN_γ_), WJ-MSC can also mediate T_reg_ cells (regulatory T cells). This helps in the polarization procedure of monocytes/macrophages toward an anti-inflammatory phenotype (type 2) and blocks the separation into the phenotype (type 1) and DCs. More specifically, the MSC-secreted interleukin 1 receptor antagonist (IL1-RA) helps in the polarization procedure of macrophages toward phenotype (type 2). Along with these, proinflammatory Th1 is switched to anti-inflammatory Th2 cells, and the profiles of cytokines are modified. It enhances IL-4 secretion by Th2 cells, increases the production of T_reg_, and blocks the multiplication, stimulation, and cytotoxicity of natural killer cells (NK cells). It directly blocks the multiplication of alloreactive CD4^+^ and CD8^+^ T cells in the absence of other immune cells, which are stimulated by the SC-derived galectin-1. Effector CD4^+^ T cells can be easily converted to Foxp3^+^ T_regs_ by B_regs_ (regulatory B cells), which produce IL-10. Along with T cells, it also blocks the multiplication of B cells and T-cell-secreted IFN_γ_. In this way, the upregulation of genes involved in the process of phagocytosis in macrophages is increased along with the downregulation of inflammatory cytokines via macrophages.

#### 2.4.3. Clinical Trial, Route of Administration, Dose, and Dosage Form

Assistance Publique–Hôpitaux de Paris had already completed a phase 1/2, randomized, triple-blinded, and parallel-assigned clinical trial (NCT04333368) in 47 adults and older adults [54]. The intervention was administered through the central venous line for 1 h by tubing and at a dose of 1 mg/kg. The placebo group was given 0.9% NaCl through the IV infusion route. The difference in ratio of partial pressure of oxygen to fractional inspired oxygen (PaO_2_/FiO_2_) was not significant between the intervention and placebo group (95% CI) [55]. Adverse events were observed in 28.6% participants from the intervention group and 25% patients from the placebo group. Efficacy of WJ-MSC was also evaluated for the treatment of COVID-19 related acute respiratory distress syndrome (ARDS) [56]. However, sponsors did not disclose results for this clinical trial (NCT04625738).

#### 2.4.4. Regulatory Approval and Marketing Authorization

WJ-MSC is not approved by USFDA and EMA so far for use against COVID-19.

### 2.5. Convalescent Plasma (CP)

#### 2.5.1. General Description

CP is the plasma that is collected from patients who have survived from any viral disease [57]. CP contains antibodies against a virus. COVID-19 survivors have antibodies against SARS-CoV-2 in their blood, and these antibodies have the potential to treat COVID-19 patients. Collecting CP from patients surviving from COVID-19 infection and administering it to patients suffering from this infection may boost the immunity of COVID-19 patients. Although CP therapy is not approved yet, USFDA proclaimed that CP therapy can be used for the treatment of critically ill COVID-19 patients.

#### 2.5.2. Mechanism of Action

In COVID-19 patients, some issues were observed, such as infiltration of inflammatory cells and cytokine storm in the alveoli of the lungs, which further produces ARDS [58,59]. Moreover, the amount of lymphocytes and cytokine levels are also reduced; on the other hand, the levels of IL-6, TNF-α, IL-10, colony-stimulating factor, and granulocyte macrophage are enhanced significantly. CP administration in COVID-19 patients enhances the lymphocyte amount and oxygen saturation and restores liver function. Therefore, CP therapy can boost immunogenicity and reduce the inflammation of lungs. However, this therapy also treats different hemorrhagic fevers induced by Ebola, influenza A (H5N1), SARS, Middle East respiratory syndrome (MERS), and other viruses [60].

#### 2.5.3. Clinical Trial, Route of Administration, Dose, and Dosage Form

An open-label study (NCT04321421) was conducted by Foundation IRCCS San Matteo Hospital in collaboration with Ospedale Carlo Poma Asst Mantova, Ospedale Maggiore Lodi, and Ospedale Asst Cremona in 49 adults and older adults having moderate to severe ARDS, which was lasting <10 days, at Catherine Klersy, Italy [61]. Participants were administered 250–300 mL CP three times over 5 days through IV infusion. However, in this study, the developers were uncertain whether CP was beneficial against COVID-19 or not [62,63]. A different study was conducted by Mayo Clinic (NCT04338360) in adults and older adults at four places in the US, namely, Arizona, Florida, Minnesota, and Wisconsin [64]. Serious adverse events were significantly reduced (<1%), and the mortality rate, up to 30 days of study, was 25.2% (95% CI) [65]. The study was successful, and the developers obtained marketing approval.

#### 2.5.4. Regulatory Approval and Marketing Authorization

Though USFDA has not approved CP therapy as yet, an IND application for it proclaims that CP can be used effectively in emergency cases and in critically ill patients [66].

### 2.6. Sarilumab

#### 2.6.1. General Description

Sarilumab is a potent MAb (IgG_1_) that inhibits IL-6 receptors [67,68]. It is approved by FDA and EMA for the treatment of rheumatoid arthritis through SC administration. In COVID-19 infection, this agent is thought to be a potent agent because of the evidence of its in vitro and in vivo therapeutic efficacy.

#### 2.6.2. Mechanism of Action

IL-6 is considered to be the pleiotropic cytokine that is responsible for the stimulation of T cells and B cells [67,68,69,70,71]. It is noteworthy that hepatocytes are also stimulated by IL-6 and CRP, which secrete fibrinogen and serum amyloid A (SAA). IL-6 also plays a pivotal role in the process of pathological inflammation. Sarilumab binds to the IL-6 receptor (both soluble IL-6 receptor, namely, sIL-6Rα, and membrane-bound IL-6 receptor, namely, mIL-6Rα). It blocks the gp130 and signal transducer and signaling transcription protein 3 (STAT3) pathway and cis- and trans- signaling pathways (though in vitro). Complexes of IL-6 and mIL-6Rα inducing trans-signaling pathways are also inhibited in those respective cells, which express specifically gp130 and mIL-6Rα.

#### 2.6.3. Clinical Trial, Route of Administration, Dose, and Dosage Form

A randomized, double-blind, and placebo-controlled phase 2/3 clinical trial (NCT04315298) was conducted at 62 locations in the US by Regeneron Pharmaceuticals in collaboration with Sanofi in 1912 adult or older adult patients having confirmed COVID-19 infection and hospitalized with pneumonia or multisystem organ dysfunctions [72]. The trial was supported by USFDA and the Biomedical Advanced Research and Development Authority (BARDA) [73,74]. In the phase 2 clinical trial, the first two experimental groups were administered 200 and 400 mg of sarilumab through the IV route [75]. The next four groups were divided into three cohorts, which received 200 mg (Cohort 1), 400 mg (Cohort 1), 800 mg (Cohort 2), and 800 mg (Cohort 3) of the drug through the IV route. It was observed (especially in phase 3) that the percentage of improvement was 43.2% for the interventional group and 35.5% for the placebo group. Another phase 3, randomized, quadruple-blinded, parallel-mode (NCT04327388) trial, by Sanofi in collaboration with Regeneron Pharmaceuticals, in 420 adults or older adults was conducted at 47 different locations in Brazil, Argentina, Chile, Canada, Germany, France, Japan, Israel, Russian Federation, Italy, and Spain [76]. The participants were divided into three groups, of which the first experimental group was administered a single dose of 200 mg sarilumab through IV injection once a day on the first day. The second group was administered a single dose of 400 mg of sarilumab through IV injection once a day. The comparator group was provided placebo through IV injection once a day on the first day. However, no significant difference in therapeutic activity was observed between groups receiving 200 mg sarilumab (*n* = 159 (38%)), 400 mg sarilumab (*n* = 173 (42%)), and the placebo (*n* = 84 (24%)) [77]. Adverse events were 70% with 400 mg sarilumab, 65% with 200 mg sarilumab, and 65% in the placebo group.

### 2.7. Tocilizumab

#### 2.7.1. General Description

Tocilizumab is a recombinant MAb that contains human and murine components [78,79]. Antigen-binding domains of the murine antihuman IL-6R antibody are grafted to human IgG_1_ scaffolding. It is effective against inflammation and autoimmune diseases by blocking IL-6R. This MAb is thought to be a potential agent to combat against COVID-19. Tocilizumab is already available in the market as Actemra^®^ and ACTPen™. Actemra^®^ is single-dose manual injection of tocilizumab, and ACTPen™ is a single-dose autoinjector containing tocilizumab [80,81]. Actemra^®^ and ACTPen™ can be administered via the subcutaneous and IV infusion route. Hoffmann-La Roche marketed these products for the treatment of arthritis [82].

#### 2.7.2. Mechanism of Action

Like sarilumab, tocilizumab competitively blocks IL-6R receptors, both sIL-6R and mIL-6R, resulting in the inhibition of signal transduction, leading to the reduction in inflammation [78]. B cells and T cells generate IL-6, which is a proinflammatory cytokine and helps in the formation of an antibody and differentiation of cytotoxic T cells and is responsible for the stimulation of hepatocytes, CRP, fibrinogen, and SAA secretion along with the diminishing of the differentiation of T_reg_ cells. It is conventionally used for the treatment of rheumatoid arthritis, systemic juvenile idiopathic arthritis, giant cell arteritis, polyarticular juvenile idiopathic arthritis, cytokine release syndrome, and so on.

#### 2.7.3. Clinical Trial, Route of Administration, Dose, and Dosage Form

Another phase 3, randomized, double-blinded, parallel-mode, and multicenter clinical trial (NCT04320615) was conducted by Hoffmann-La Roche in 452 adult or older adult patients at 62 different locations in the U.S., Canada, Denmark, Germany, France, Italy, Netherlands, Spain, and UK [83]. Participants were administered 8 mg/kg of this drug (maximum up to 800 mg each dose), and one more dose was only administered in case of worse condition or lack of improvement through IV infusion. It was observed that 24.5% of the participants from the interventional group and 25% of the participants from the placebo group died by the 60th day [84]. Adverse events were seen in 24.1% of the participants (interventional group) and 29.4% of the participants (placebo group). Result indicates that no beneficial outcomes were obtained in this study.

#### 2.7.4. Regulatory Approval and Marketing Authorization

Actemra^®^ (tocilizumab), manufactured by Genentech (member of the Roche Group) is approved by USFDA in December 2022 for severe COVID-19 patients [85].

### 2.8. Bevacizumab

#### 2.8.1. General Description

Like tocilizumab, bevacizumab is also a humanized IgG_1_ MAb that has the ability to bind with vascular endothelial growth factor A (VEGF), resulting in the apoptosis of tumor cells [86]. It is conventionally used for the treatment of various types of cancers, such as cervical cancer, colorectal cancer, non-small cell lung cancer, fallopian tube cancer, ovarian cancer, peritoneal cancer, and renal cell carcinoma. It is already a FDA-approved agent for the treatment of cancer [87,88]. Currently, it is being evaluated for its potential efficacy against COVID-19 infection for which many clinical trials are being continued. Bevacizumab is already available in the market as Avastin^®^, Mvasi^®^, and Zirabev^®^. Avastin^®^ injections are used for the treatment of diseases such as colorectal cancer and non-small cell lung cancer [86,87].

#### 2.8.2. Mechanism of Action

Bevacizumab binds to VEGF, a well-known proangiogenic growth factor expressed by tumor cells, and blocks VEGF receptors of lung tissues, leading to an increase in vascular permeability [87,88,89]. Therefore, oxygen perfusion and anti-inflammatory effects increase.

#### 2.8.3. Clinical Trial, Route of Administration, Dose and Dosage Form

Another phase 2 open-label trial (NCT04275414), in 27 adults or older adults having confirmed COVID-19 infection, was conducted by Qilu Hospital of Shandong University in collaboration with Renmin Hospital of Wuhan University and Moriggia-Pelascini Gravedona Hospital, Gravedona, Italy [90]. Patients were administered with 500 mg of bevacizumab along with 100 mL normal saline through IV drip for a minimum of 90 min and onwards. Body temperature was normalized in 93% patients and improvement in PaO_2_/FiO_2_ ratios was observed in 92% patients who received both bevacizumab and the standard of care [89].

#### 2.8.4. Regulatory Approval and Marketing Authorization

Till date, bevacizumab is not approved for the treatment of COVID-19.

## 3. Comparative Analysis of Newly Discovered Molecules

There are several differences among new molecules in different aspects, such as dose, dosage forms, duration of administration, efficacy, and adverse events. Molnupiravir was found effective at higher doses when compared with others (400 and 800 mg BID oral dose) [14,21,22,24]. Although only 6.8% death or hospitalization case was reported for molnupiravir, the mutation of the patient’s DNA is a significant adverse effect [91]. On the other hand, only a negligible difference was observed between molnupiravir and placebo for other adverse events. A recent study showed that on day 5, 18.42% of patients treated with molnupiravir survived from COVID-19 (especially caused by the omicron variant), while in the placebo group, none survived [92]. Adverse events were reported in 3.9% of COVID-19 patients who were treated with molnupiravir. Unlike molnupiravir, Paxlovid™ did not cause any mutation in host DNA [35]. Molnupiravir inhibits RdRp to block the viral transcription, while Paxlovid™ (especially nirmatrelvir) is a protease inhibitor and acts on the 3-chymotrypsin-like cysteine protease enzyme (M^pro^) [91]. Pharmacokinetic parameters of nirmatrelvir increases when administered with ritonavir due to CYP3A4 inhibition, leading to a reduction in the metabolism of nirmatrelvir [93]. In an EPIC-PEP study, molnupiravir showed a superior efficacy of 89% reduction in hospitalization or death cases [33]. On the other hand, 1.7% of patients from the Paxlovid™ group showed adverse events, while 6.6% of patients reported adverse effects from placebo.

Besides other novel molecules, novel MAbs have also shown beneficial effects in COVID-19. Baricitinib showed a higher mortality rate (38.2%) when compared with molnupiravir (6.8%) and Paxlovid™ (11%) [42]. More trials on baricitinib are required to be conducted in a wide population pool to have a true picture on adverse events. It may furnish a more clear result. Moreover, 15% of patients from the baricitinib group reported adverse effects, whereas adverse effects for molnupiravir and Paxlovid™ were reported by 3.9% and 1.7% of patients, respectively. Sarilumab is administered through the IV route, while other drugs require oral administration. However, improvements were observed in 43.2% of patients with sarilumab and 35.5% of patients reported improvement with placebo administration. These results clearly indicate that molnupiravir and Paxlovid™ have superior efficacy than sarilumab. Moreover, higher adverse effects were observed with sarilumab (70% of patients from 400 mg and 65% of patients from 200 mg) in comparison with molnupiravir and Paxlovid™ [77]. In the clinical trial, 24.5% of patients treated with tocilizumab were found dead on day 60 [84]. This result was better than sarilumab (56.8% patients), but worse than molnupiravir (6.8% of patients), Paxlovid™ (11% of patients), and baricitinib (38.2% of patients). Besides, adverse events were observed in 24.1% of patients with tocilizumab. Contrary to this, fewer patients reported adverse effects when treated with molnupiravir (3.9% of patients), Paxlovid™ (1.7% of patients), and baricitinib (15% of patients). The condition of 92% of patients improved with the IV administration of bevacizumab [89]. Bevacizumab normalized the body temperature of patients within a very short time. Unfortunately, adverse events observed for bevacizumab were high, almost 38%.

WJ-MSC and CP therapy were used in hospitalized patients and emergency conditions, but clinical trials have not yet ascertained their therapeutic efficacy. WJ-MSCs work on mitogen-induced T cells, and CP works on the cytokine storm to provide protection for patients from COVID-19 [50,51,52,53,58,59]. However, both therapies require IV administration. For WJ-MSC therapy, adverse events were seen in 28.6% of patients, and for CP therapy, it was less than 1% [55,65]. On the basis of adverse events, it can be proclaimed that CP therapy is well tolerated by patients. This is probably because CP has been obtained from patients who have survived COVID-19. Figure 1 and Figure 2 describe the percentage of survival rates and adverse events, respectively, for these new molecules. Molnupiravir and Paxlovid™ are better in terms of survival rate and adverse events.

Drugs such as sarilumab and tocilizumab target cytokines such as IL-6 for reducing inflammation. Some drugs inhibit RNA replication either by blocking RdRp (such as molnupiravir) or 3CL (such as Paxlovid™). The rest of the drugs block the entry of the virus either by the JAK-STAT pathway (like sarilumab and baricitinib) or by blocking the NAK family (such as baricitinib). However, the blocking of inflammation can lead to the alleviation of COVID-19 symptoms (such as pneumonia). Therefore, targeting cytokines for blocking inflammation may be a therapeutic management of COVID-19. MOAs of new molecules are illustrated in Figure 3.

Table 1 summarizes some clinical trials conducted on new molecules and materials for the treatment of COVID-19.

## 4. Repurposed Drugs

Several repurposed drugs, which had received approval from drug regulatory authorities for the treatment of diseases other than COVID-19, have been tested clinically against COVID-19. Several clinical trials on pre-existing drugs for the treatment of COVID-19 are still ongoing.

### 4.1. Dexamethasone

#### 4.1.1. General Description

Dexamethasone is a corticosteroid that blocks the release of inflammation causing compounds [94]. For many years, it has been widely used for the treatment of hematologic, endocrine, respiratory, dermatologic, allergic, collagen, gastrointestinal, ophthalmic, neoplastic, rheumatic, edematous, and other conditions, more specifically breathing disorders, ulcerative colitis, psoriasis, systemic lupus erythematosus, and so on [94]. Generally, it cannot be used in patients having fungal infection as it weakens the immune system of the body. It has potent anti-inflammatory activity, which facilitates it for assessing through different clinical trials in COVID-19 patients.

#### 4.1.2. Mechanism of Action

Dexamethasone blocks the demargination and apoptosis of neutrophil and phospholipase A2, resulting in the reduction in the production of the derivatives of arachidonic acid [94,95]. It also blocks nuclear factor κB (NFκB) and proinflammatory cytokines (IL-1, IL-6, IL-2, IL-8, TNF, VEGF, IFN_γ_, and prostaglandin), which are responsible for the transcription of inflammation and also helps in the activation of anti-inflammatory genes.

#### 4.1.3. Clinical Trial, Route of Administration, Dose, and Dosage Form

A phase 3, randomized, open-label, parallel-mode (NCT04327401) clinical trial of dexamethasone was conducted in 290 patients at 21 places of Brazil by the Hospital Israelita Albert Einstein, Hospital do Coracao, Brazilian Research in Intensive Care Network and Ache Laboratorios Farmaceuticos S.A. [96]. Patients were administered dexamethasone 20 mg once in a day for 5 days, followed by 10 mg once in a day for 5 days through an intravenous route. Unfortunately, the trial was terminated by the data monitoring committee for the unaccepted recovery results in trial. Dr. Negrin University Hospital in collaboration with the Li Ka Shing Knowledge Institute, Consorcio Centro de Investigación Biomédica en Red, M.P., conducted a phase 4, randomized, open-label, parallel-mode (NCT04325061) clinical trial on dexamethasone in 200 adult or older adult patients at 24 different places in Spain [97]. Patients received 20 mg dexamethasone dose daily up to 5 days randomly and, after that, 10 mg dose daily up to 5 days randomly through the IV route. This trial was also terminated due to lack of participant enrollment.

#### 4.1.4. Regulatory Approval

Dexamethasone is not approved till date for the treatment of COVID-19. However, WHO recommended dexamethasone for the treatment of severe COVID-19 patients in October 2020 [98].

### 4.2. Naproxen

#### 4.2.1. General Description

Naproxen is an over-the-counter (OTC) nonsteroidal anti-inflammatory drug (NSAID) indicated for the treatment of acute pain and pain involved in rheumatoid arthritis and other diseases, such as headache, tendonitis, muscle pain, menstrual pain, dental pain, swelling, and joint stiffness [99,100]. It is also available in the market alone and in combination with esomeprazole (for gastric ulcers) or with sumatriptan (for migraine).

#### 4.2.2. Mechanism of Action

Naproxen is a nonselective cyclooxygenase (COX) inhibitor that inhibits arachidonate (ester form of arachidonic acid), leading to its anti-inflammatory and analgesic activities [100,101]. Naproxen blocks the nucleoprotein of SARS-CoV-2, resulting in blocking the RNA replication of the virus [102].

#### 4.2.3. Clinical Trial, Route of Administration, Dose, and Dosage Form

Assistance Publique—Hôpitaux de Paris conducted a phase 3, open-label, randomized, parallel model (NCT04325633) clinical trial on naproxen in 584 adult or older adult patients [103]. Naproxen was orally administered at 250 mg twice a day along with 30 mg of lansoprazole. This treatment was compared with the standard of care. The location of this trial was not recorded in the clinical trial record of the U.S. However, this trial was terminated due to insufficient recruitment of participants.

#### 4.2.4. Regulatory Approval

No regulatory authorities have approved nor recommended naproxen till date for the treatment of COVID-19.

### 4.3. Remdesivir

#### 4.3.1. General Description

Remdesivir is a 1′-cyano-substituted nucleotide analogue and phosphoramidate prodrug. It is well known for its broad-spectrum effect against Ebola, SARS-CoV, MERS-CoV, Nipah virus, and Hendra virus [104,105]. An in vitro study of this drug proved it as a therapeutically potent agent against SARS-CoV-2 and safe for humans.

#### 4.3.2. Mechanism of Action

Remdesivir acts as a nucleoside analog and is a RdRp blocker that attacks the replication cycle of a viral genome [105,106,107,108]. CoVs utilize RdRp for running their replication process of RNA-dependent genomes. The drug is metabolized in the human body, and the metabolite GS-441524 blocks RdRp, leading to a gradual stopping of RNA formation.

#### 4.3.3. Clinical Trial, Route of Administration, Dose, and Dosage Form

A phase 3, randomized, open-label, parallel-mode clinical trial (NCT04292899) was conducted in 6000 hospitalized child, adult, or older adult patients by Gilead Sciences at 179 places in the US, France, Japan, China, Hong Kong, Italy, Republic of Korea, Taiwan, Germany, Netherlands, Singapore, Sweden, Spain, Switzerland, and UK [109]. In this trial, the patients were divided into three groups. The first group received 200 mg drug for 5 days and subsequently 100 mg drug for up to 5 days through IV infusion. The second group was administered with 200 mg drug for 10 days and then 100 mg drug for up to 10 days through IV infusion. The third group received 200 mg drug for 5 or 10 days and then 100 mg drug for up to 5 or 10 days through IV infusion. After 14 days, 74.4% of patients from interventional groups and 59% of patients from noninterventional groups survived (95% CI; *p* < 0.001) [110]. The National Institute of Allergy and Infectious Diseases (NIAID) conducted a phase 3, randomized, double-blinded, and parallel-mode (NCT04280705) clinical trial in 572 hospitalized adult or older adult patients at 68 different places [111]. In this study, 200 mg of remdesivir was administered to patients through the IV route along with a maintenance dose of 100 mg once a day during hospitalization for 10 days, and its efficacy was compared with placebo against COVID-19 infection. It was observed that the median recovery time was 10 days and 15 days for the intervention and placebo group, respectively [112]. Serious adverse events were seen in 24.6% of patients in the interventional group and 31.6% of patients treated with placebo.

#### 4.3.4. Regulatory Approval

Veklury^®^ (remdesivir) IV infusion, developed by Gilead Sciences, obtained EUA from USFDA in October 2020 and obtained expedited approval from USFDA on January 2022 [113,114]. Veklury^®^ is approved for the treatment of hospitalized adult and pediatric patients (at least having 40 kg body weight) who are at high risk of COVID-19. EMA granted full approval for Veklury^®^ (remdesivir) IV infusion in December 2021 [115].

### 4.4. Hydroxychloroquine

#### 4.4.1. General Description

Hydroxychloroquine (chemically 4-aminoquinoline) is a well-known derivative of chloroquine [116,117]. Apart from its antimalarial activity, it has proven effect as anti-inflammatory, antiautophagy, and immunosuppressive activities for the treatment of systemic lupus erythematosus and rheumatoid arthritis. It is also used as a chemotherapeutic drug against malarial parasites.

#### 4.4.2. Mechanism of Action

Hydroxychloroquine blocks sialic acid receptors at the upper respiratory pathway and prevents the glycosylation of the angiotensin converting enzyme 2 (ACE2) receptor [116,117,118,119]. Prevention of ACE2 receptor glycosylation results in the failure of the viral spike protein binding to the ACE2 receptor. Therefore, the entry of the COVID-19 virus in humans was restricted. Hydroxychloroquine also blocks antigen processing and lowers the T-cell level in the human body. This results in the prevention of cytokine storm.

#### 4.4.3. Clinical Trial, Route of Administration, Dose, and Dosage Form

Several clinical trials, NCT04341493, NCT04342156, NCT04341441, NCT04325893, and NCT04329923, were terminated due to reasons such as continuous decreasing number of participants, increasing side effects, or lack of beneficial results. A phase 3, randomized, quadruple-blinded, parallel-mode clinical trial (NCT04332991) was conducted in 510 adult or older adult patients at 43 locations in the US by Massachusetts General Hospital in collaboration with the National Heart, Lung, and Blood Institute (NHLBI) [120,121]. Participants were divided into the experimental group and comparator group, of which the experimental group was administered 400 mg hydroxychloroquine sulfate BID on the first day and then 200 mg of this drug BID for the rest of 4 days through the oral route (tablet), whereas the comparator group was administered placebo through the oral route [122]. By day 28, it was observed that 10.4% of participants from the interventional group and 10.6% of participants from the placebo group had died [123]. The result of the trial indicates that there were no significant differences between the interventional group and the placebo group.

#### 4.4.4. Regulatory Approval

Hydroxychloroquine is approved for treating emergency cases of COVID-19 infection and for the prophylaxis of healthcare professionals by FDA and EMA [116,124,125]. USFDA granted EUA to hydroxychloroquine for the emergency treatment of COVID-19 [126,127]. EMA released news on 1 April 2020 stating that chloroquine and hydroxychloroquine can be used for the emergency condition of a patient and in clinical trials [128].

### 4.5. Favipiravir

#### 4.5.1. General Description

The drug favipiravir (also known as Avigan or Favilavir) is a derivative of pyrazine and is developed by Fujifilm Toyama Chemical Co., Ltd. of Tokyo, Japan [129,130]. It was approved for the treatment of influenza A and B as it attacks RdRp, required for the replication cycle and transcription process of the viral genome. Its therapeutic efficacy against Lassa virus and Ebola virus is under investigation.

#### 4.5.2. Mechanism of Action

Favipiravir is a prodrug that is biotransformed into its active metabolite favipiravir ribofuranosyl-5′-triphosphate (favipiravir-RTP). Favipiravir-RTP interacts with RdRp and blocks the replication cycle of the virus genome [78,118,131]. Till now, the interaction mechanism is not very clear, but a hypothesis regarding it claims that this active metabolite may link to the conserved domains of polymerase, or it may be added in a nascent viral RNA, for which nucleotides cannot be added, and the process of transcription and replication is blocked. To accomplish the incorporation, this active metabolite has to compete with the purine bases or nucleotides.

#### 4.5.3. Clinical Trial, Route of Administration, Dose, and Dosage Form

A phase 2, randomized, open-label, parallel-mode clinical trial (NCT04346628) was conducted in 120 adult or older adult patients by Stanford University [132]. Participants were split into two sections, of which the experimental section received 1800 mg favipiravir for the first day along with 800 mg twice a day for up to 10 days through the oral route. However, no significant differences (95% CI) were observed between the experimental group and the comparator group for mortality, symptom resolution, and transition as well as accumulation of mutation in the viral genome [133]. Another phase 3, randomized, open-label, parallel-mode clinical trial (NCT04349241) was conducted by Ain Shams University in 100 adult or older adult patients at the Faculty of Medicine, Ain Shams University Research Institute–Clinical Research Center, Cairo, Egypt [134]. The experimental group was administered 1600 mg of favipiravir BID every 12 h on the 1st day and 600 mg of this drug BID every 12 h as the maintenance dose from the 2nd day to the 10th day through the oral route, and the comparator group received the standard of care according to the national Egyptian guideline, that is, 75 mg of oseltamivir every 12 h from the 5th day to the 10th day and 400 mg of hydroxychloroquine every 12 h on the 1st day and 200 mg of this drug every 12 h daily from the 2nd day to the 10th day through the oral route. Among all groups, the favipiravir group showed potential results, where two consecutive SARS-CoV-2-negative reports were observed after the treatment and body temperature was also normalized [135,136].

#### 4.5.4. Regulatory Approval

DCGI (Drug Controller General of India) granted EUA to favipiravir for the treatment of severe COVID-19 patients on July 2020 [137]. It is not approved by USFDA and EMA till date.

### 4.6. Umifenovir (Arbidol)

#### 4.6.1. General Description

Umifenovir is a derivative of the indole moiety. In Russia, it is an approved drug to treat influenza A and B [78,138,139]. This antiviral drug is effective against viruses such as the Lassa virus, coxsackie virus B5, herpes simplex virus, hepatitis B and C, Zika virus, Ebola virus, and chikungunya virus.

#### 4.6.2. Mechanism of Action

Umifenovir can interact with amino acid residues such as tryptophan and tyrosine and thereby aids in preventing the entry of viruses in the human body [140]. It also interacts with aromatic residues present in glycoproteins of a virus (S protein of SARS-CoV-2). The glycoprotein is generally responsible for the recognition for cells and the process of fusion with the plasma membrane. Through the clathrin-mediated exocytosis, the process and trafficking in the cells are blocked. In enveloped viruses, such as SARS-CoV-2, it can also directly bind with the lipid envelope, leading to the stabilization of the plasma membrane, which prevents the entry of the virus in the human body.

#### 4.6.3. Clinical Trial, Route of Administration, Dose, and Dosage Form

Another phase 4, randomized, open-label, parallel-mode clinical trial (NCT04252885) was conducted by Guangzhou Eighth People’s Hospital in 125 adult or older adult patients at Guangzhou Eighth People’s Hospital, Guangdong, China [141]. Participants were divided into three groups. One group was administered 200 mg of lopinavir oral tablet with 50 mg of ritonavir oral tablet twice a day for 7 to 14 days along with the standard treatment. Another group received 200 mg of umifenovir (i.e., two tablets of 100 mg of arbidol) thrice in a day for 7 to 14 days along with the standard treatment. The third group was not provided any intervention. On the seventh day, 23.5% of participants from the lopinavir/ritonavir group, 8.6% of participants from the arbidol group, and 11.8% of participants from the control group showed deterioration from moderate to severe condition (*p* = 0.206) [142]. Adverse events were observed in 35.3% of patients treated with lopinavir/ritonavir and 14.3% of patients treated with arbidol. Adverse events were not observed in patients who received placebo. Result indicates that arbidol monotherapy could be beneficial for mild, moderate, and severe COVID-19 patients.

#### 4.6.4. Regulatory Approval

Umifenovir is not approved by USFDA and EMA for the treatment of COVID-19 till date.

### 4.7. Azvudine

#### 4.7.1. General Description

Azvudine (2′-deoxy-2′-β-fluoro-4′-azidocytidine), also known as FNC, is a prodrug that is biotransformed to its active form, FNC triphosphate. Azvudine is the first oral antiviral drug approved in China by SFDA [143]. It is a nucleoside analogue, having broad spectrum antiviral efficacy against RNA viruses such as Ebola, HIV, HBV, HCV, and so on [144]. It contains 4′-modified nucleoside, which is essential for drug resistance and antiviral effect [143]. It is notable that azvudine, manufactured by Zhengzhou Granlen PharmaTech Ltd. (a division of Granlen LLC, Carlsbad, CA, USA), received approval from the China Food and Drug Administration (SFDA) in April 2013 as an investigational new drug for the treatment of HIV [145]. Azvudine, manufactured by Genuine Biotech Limited, Pingdingshan, Henan Province, Central China, is available in the Chinese market as 1 mg oral tablet for the treatment of COVID-19 [146].

#### 4.7.2. Mechanism of Action

FNC works against SARS-CoV-2 by blocking RdRp [144,147]. The mechanism or action is divided into two steps, antiviral action in the thymus (one of the vital immune organs, where T lymphocytes are born) of the human chest and boosting immunity against the virus in whole body. After distribution, FNC is phosphorylated three times in the cytoplasm by deoxycytidine kinase and forms the active product FNC triphosphate, which inhibits RdRp of SARS-CoV-2 in the thymus part. However, after inhibition of RdRp, the FNC triphosphate elevates IL-4, IL-10, and IL-13 levels in the thymus to combat cytokine storm. FNC can also control the expression of P-glycoprotein (Pgp), multidrug resistance-associated protein 2 (MRP2), breast cancer resistance protein (BCRP), and so on [148].

#### 4.7.3. Clinical Trial, Route of Administration, Dose, and Dosage Form

A randomized, open-label, controlled pilot study (phase 0, ChiCTR2000029853) on FNC was conducted in 20 mild to moderate COVID-19 patients at People’s Hospital of Guangshan County, Guangshan County, He’nan, China [147]. Twelve patients (60%) were provided routine treatment. Among these 12 patients, 6 patients were administered placebo, and the other 6 received FNC (5 mg per day and 5 oral tablets QD). After that, some patients were provided antiviral treatment, some received antibiotic treatment, some patients received adjuvant treatment, and some were administered traditional Chinese medicine. As a result, the placebo group showed adverse events with 30% incidence (*p* = 0.06), while the FNC group did not show any adverse event. After 5 days of treatment with FNC, 100% of patients survived from COVID-19 pneumonia, while in the case of the placebo group, 100% survival was observed after 8 days. FNC was also found effective in those patients who received other treatments (antiviral, antibiotic treatment, etc.) [149].

#### 4.7.4. Regulatory Approval

In July 2022, azvudine obtained conditional approval from the National Medical Products Administration, China [143]. On the other hand, azvudine obtained approval from the National Health Commission and National Administration of Traditional Chinese Medicine, China, and the National Healthcare Security Administration, China, for incorporating it in the Diagnosis and Treatment Program for Novel Coronavirus Pneumonia (Ninth Edition) and a medical reimbursement list as a potential drug for the treatment of COVID-19 in August 2022.

## 5. Comparative Analysis of Repurposed Drugs

Repurposed drugs have been symptomatically used for treating inflammation, pneumonia, and other symptoms observed in COVID-19 patients. Dexamethasone has been used due to its efficacy against inflammation [94]. According to one study, dexamethasone has reduced the risk of death by 1/3 in patients kept on ventilation and by 1/5 among patients requiring oxygen [150,151]. Despite having advantages, such as low cost, prolonged action, and so on, dexamethasone reduced immunity in COVID-19 patients. Dexamethasone also causes other side effects, such as anxiety and disturbance in sleep hormonal imbalance. It also causes some withdrawal symptoms, such as myalgia, arthralgia, vomiting, and loss of weight headache. In an open-level, randomized clinical trial (NCT04381936; RECOVERY trial), 6 mg of dexamethasone (QD up to 10 days) was administered to hospitalized COVID-19 patients through the oral or IV route [152]. In that trial, 2104 patients received dexamethasone and 4321 patients received standard care. It was observed that 22.9% of patients from the dexamethasone group and 25.7% of patients from the standard of care group died within 28 days (95% CI, *p* < 0.001). Results indicated a marginal reduction in death rate. Another study reported that patients treated with dexamethasone showed 12.7% adverse events [153].

Naproxen, a nonselective COX inhibitor, such as dexamethasone, also inhibits inflammation to combat COVID-19 [100,101]. However, in a randomized placebo controlled, double-blinded clinical trial, naproxen oral capsule (500 mg BID) showed potential efficacy in treating breathing issue and cough in COVID-19 patients [154]. It also elevated mean the corpuscular volume and helped to normalize blood pressure. In overall aspects, naproxen has proven itself more beneficial than dexamethasone. In general, treatment with NSAIDs may cause gastrointestinal issues, but naproxen did not show any serious adverse event. Naproxen is therefore better tolerated when compared with dexamethasone.

1′-Cyano-substituted nucleotide analogue remdesivir has shown both positive and negative effects in COVID-19 patients [104,105]. Like molnupiravir, remdesivir blocks RdRp to inhibit RNA formation in SARS-CoV-2 [105,106,107]. In a clinical trial (NCT04292899), it was observed that the survival rate of patients treated with iv infusion of remdesivir (IV infusion) was 74.4% (95% CI; *p* < 0.001) [110]. For the IV route, the death rate was recorded at 7.6% (95% CI; *p* = 0.001). Results gave a clear indication that treatment though the IV route for remdesivir was better than the infusion route. Despite having a good survival rate, remdesivir also showed different adverse effects, such as reaction at the infusion site, kidney injuries, anemia, multiorgan failure, pyrexia, hyperglycemia, hypokalemia, torsade de point, hypotension, cardiac arrest, bradycardia, and complete heart block= [155]. Another study indicated that remdesivir could induce cytotoxicity on cardiomyocytes [156]. However, WHO provided a conditional approval for the administration of remdesivir in hospitalized patients having severe COVID-19 [5]. Moreover, in the another study, remdesivir showed 24.6% serious adverse events [112]. Remdesivir (7.6% patients) showed more beneficial effects than dexamethasone (22.9% patients) in reducing the death rate, but adverse events for dexamethasone (12.7% of patients) were lesser than remdesivir (24.6% of patients). Like remdesivir, favipiravir also blocks RdRp to combat COVID-19 [78,118,131]. Due to its good efficacy in inhibiting viral proliferation, favipiravir protects the lungs from fibrosis and inflammatory damage [157]. In this aspect favipiravir possess more beneficial effects than remdesivir. Remdesivir may also protect the lung but has serious adverse events, leading to its cautious use for the treatment of COVID-19. Most of the clinical trials of favipiravir have been conducted in combination other drugs such as oseltamivir, darunavir, ritonavir, and hydroxychloroquine. This is probably due to the synergistic or additive effect in various symptoms of COVID-19. It has also been observed that a high dose of favipiravir may lead to serious adverse events such as elevated hepatic enzymes, tachycardia, and vomiting [158,159]. A study reported serious adverse events in 10.36% of COVID-19 patients when treated with favipiravir [160]. A different study revealed a 46.2% fatality rate for favipiravir [161]. In a nutshell, no beneficial effect has been observed in COVID-19 patients treated with favipiravir.

The antimalarial drug hydroxychloroquine reduces cytokines to give symptomatic relief in inflammation to COVID-19 patients [116,117]. In one study, 10.4% of COVID-19 patients died when they received 200–400 mg BID oral hydroxychloroquine treatment [123]. No significant difference in death rate between the hydroxychloroquine group and the placebo group (10.6%) was observed in the clinical trial. Serious adverse events from hydroxychloroquine were reported in 30% of COVID-19 patients when treated with hydroxychloroquine [162]. Hydroxychloroquine showed the highest adverse effects in COVID-19 patients when compared with other repurposed drugs. Both remdesivir and hydroxychloroquine were recommended conditionally for treating COVID-19, but reports of serious adverse events had compromised their clinical use. Later, regulatory authorities had restricted the use of both of these drugs in treating COVID-19. ICMR (India) had recommended the use of chloroquine or hydroxychloroquine for prophylaxis of COVID-19. For showing the desired therapeutic efficacy in COVID-19 patients, high doses of chloroquine or hydroxychloroquine is required, but high doses may cause serious adverse effects [6,9]. Overdose of chloroquine can cause muscle weakness, mood swings, nausea, skin irritation, headache, bleeding from the nose, pale skin, swelling, cramps, and hearing problems. On the other hand, an overdose of hydroxychloroquine can cause cramps, drowsiness, headaches, nausea, irreversible blindness, lowering of blood glucose, depression, reduced appetite, and heart failure. An overdose for both cases can cause death. Therefore, it is highly recommended not to take these drugs without the advice of a physician. A Chinese study reported hydroxychloroquine to be ineffective against COVID-19. A different study reported hydroxychloroquine to be effective when it is used in combination with azithromycin [163].

Umifenovir (arbidol) has been used for treating diseases caused by several viruses such as hepatitis B and C, Zika, and influenza [78,138,139]. Adverse events were found in 14.3% of COVID-19 patients treated with umifenovir, and therefore, it may be regarded as superior to remdesivir (24.6% of patients) and hydroxychloroquine (30% of patients) but inferior when compared with favipiravir (10.36% of patients) [142]. Notably, the dose of umifenovir is lower in comparison with other repurposed drugs. Similarity was found in the case of the route of administration, oral route. A recent study revealed the mortality rate for the umifenovir group and the control group, that is, 33.3% and 38%, respectively [164]. These results indicated no statistically significant difference in mortality rate between umifenovir and the control group. Furthermore, the mortality rate of umifenovir (33.3% of patients) was more than the death rate observed for remdesivir (7.6% of patients) and hydroxychloroquine (10.4% of patients). However, another study revealed that umifenovir has potential in reducing the mortality rate [165,166]. The first approved (SFDA) oral antiviral drug, azvudine, works like favipiravir [143]. As an RdRp inhibitor, both favipiravir and azvudine can be compared. According to the trial results, as per the survival rate of COVID-19 patients and adverse events, azvudine was superior to favipiravir [147]. Azvudine seems better among all repurposed drugs discussed here as it has the highest survival rate (100%) and adverse events are negligible. However, more clinical trials and more participants are required to prove its efficacy. Figure 4 and Figure 5 illustrate the survival rates and adverse events, respectively, for the repurposed drugs.

Some of the clinical trials on pre-existing drugs are listed in Table 2. The main therapeutic management approach for repurposed drugs was to block cytokines and therefor inhibit inflammation. To achieve this target, repurposed drugs may either directly block (like dexamethasone) cytokines or indirectly by blocking RdRp (like favipiravir). NSAIDs (such as naproxen) target COX to inhibit the cytokines’ mediated inflammation. Some of the repurposed drugs also target the replication of viral RNA (such as azvudine). Apart from that, a few drugs tried to stop the virus by blocking the ACE2 receptor or its glycosylation process (such as hydroxychloroquine). Some repurposed drugs (such as umifenovir) also work on amino acids and S-glycoprotein to prevent entry of SARS-CoV-2 in the human body. Blocking inflammation or viral RNA replication may inhibit SARS-CoV-2. MOAs of repurposed drugs are illustrated in Figure 6.

Different drugs are there in the market, which are repurposed for the treatment of COVID-19. They are approved by regulatory authorities such as USFDA, EMA, and DCGI. Some of them are listed in Table 3 along with their approved brand names.

## 6. Conclusions

Clinical studies on new discovered molecules and repurposed drugs have proven effectiveness and safety for the treatment of COVID-19. The first approved antiviral drug, molnupiravir, has shown good therapeutic efficacy, but the risk of mutation and high dose may limit its wider usage in treating COVID-19. Paxlovid™, the combination of nirmatrelvir and ritonavir, has shown good therapeutic efficacy due to the synergistic effect of nirmatrelvir and ritonavir. Ritonavir improves the pharmacokinetics of nirmatrelvir, leading to advantage in pharmacodynamics. Several studies have revealed acceptable human toleration for new discovered molecules such as Paxlovid™ and molnupiravir. MAbs have also shown potential results in clinical studies. Among all MAbs, bevacizumab has been observed better based on the survival rate of COVID-19 patients. Other MAbs (such as sarilumab, baricitinib, and tocilizumab) have scored lower due to higher mortality rate or serious adverse events. Almost all repurposed drugs, discussed here, have serious adverse events, which are dose dependent. Different clinical trials have addressed this issue to find their best dose, but still the quest to find the best dose seems unresolved. Some drugs have been initially recommended for the treatment of COVID-19, such as remdesivir and hydroxychloroquine. Unfortunately, these drugs were later found very much harmful due to their adverse effects, leading to restrictions imposed on them. Notably, most of the route of administration for most of COVID-19 drugs is oral, and very few drugs have been tested clinically for other routes, such as IV or infusion. Several drugs (such as favipiravir and umifenovir) are recommended to be administered in combination with other drugs for their synergistic effects. More clinical studies on monotherapy are required for these drugs (especially umifenovir). As per a single clinical study, it is quite impressive that azvudine had 100% survival rate and negligible adverse events. More clinical studies are required to ascertain the efficacy and safety of azvudine.

This review has some limitations, which can be addressed in future reviews. Two neutralizing antibodies, imdevimab and casirivimab, are not highlighted in this work due to lack of data on clinical trials. These antibodies can be explored in future reviews. A comparison on efficacy of the repurposed drugs against each variant of SARS-CoV-2 is not presented in this review due to lack of availability of data on clinical trials. Various clinical trials are still ongoing or for some others results are not yet available in the public domain. This limitation allows a future scope of reviews. Besides, this review has furnished details of only one study where combination drugs are used (Paxlovid™). More information on combinations of drugs used against COVID-19 can be introduced in future works.

Further work may be conducted to determine the efficacy of each drug against each variant of SARS-CoV-2. The risk of mutations in humans for other RdRp inhibitors should be determined. Adverse events from new discovered molecules seem lower when compared with repurposed drugs. However, further studies are required to prove this statement. In future studies, some new discovered drugs, such as molnupiravir and Paxlovid™, should be compared with repurposed drugs.

## Figures and Tables

**Figure 1 vaccines-11-00332-f001:**
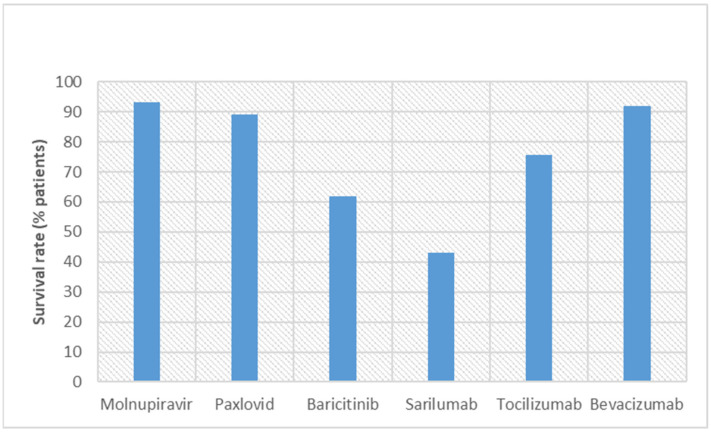
Comparison of the survival rates of COVID-19 patients in clinical trials conducted with new discovered molecules.

**Figure 2 vaccines-11-00332-f002:**
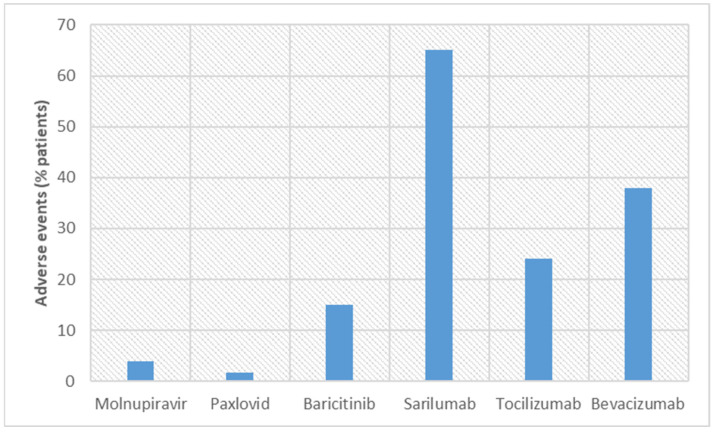
Comparison of adverse events observed in COVID-19 patients in clinical trials conducted with new discovered molecules.

**Figure 3 vaccines-11-00332-f003:**
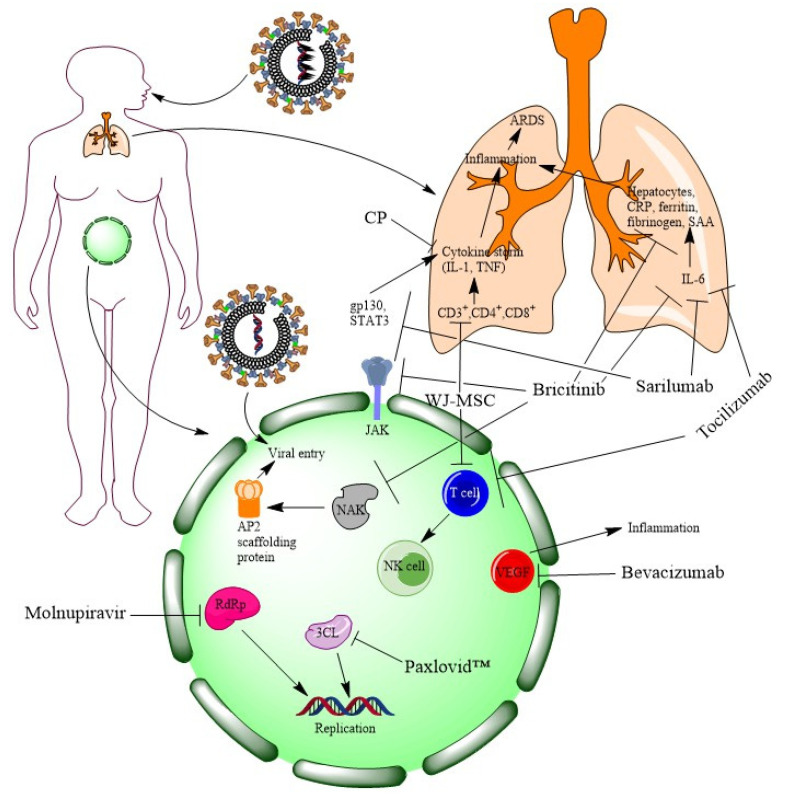
At a glance MOAs of newly discovered molecules for COVID-19 (figure is generated by ChemDraw Professional 15.1).

**Figure 4 vaccines-11-00332-f004:**
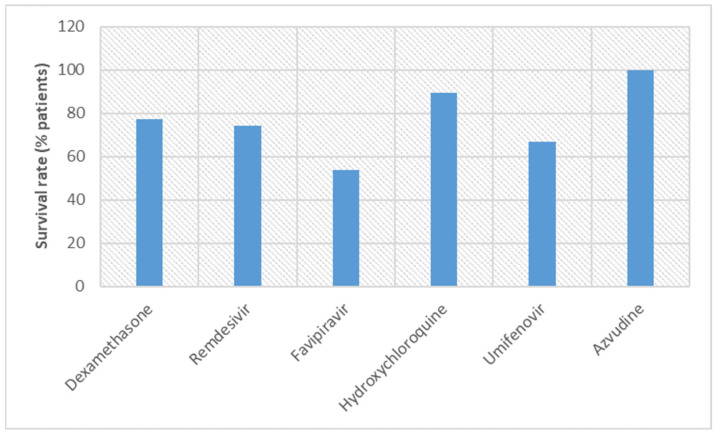
Comparison of survival rates of COVID-19 patients in clinical trials conducted with repurposed drugs.

**Figure 5 vaccines-11-00332-f005:**
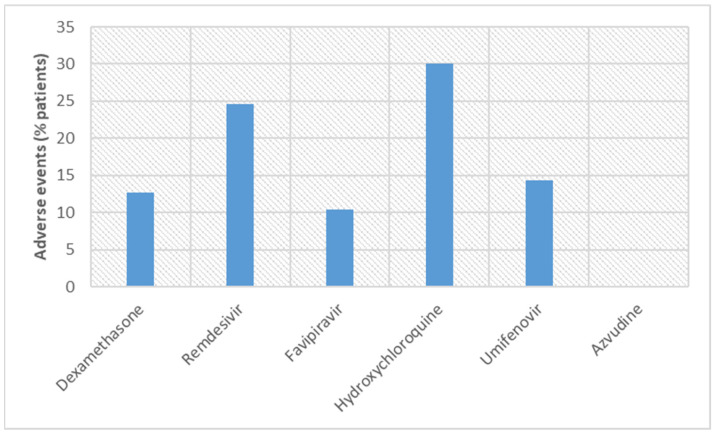
Comparison of adverse events observed in COVID-19 patients in clinical trials conducted with repurposed drugs.

**Figure 6 vaccines-11-00332-f006:**
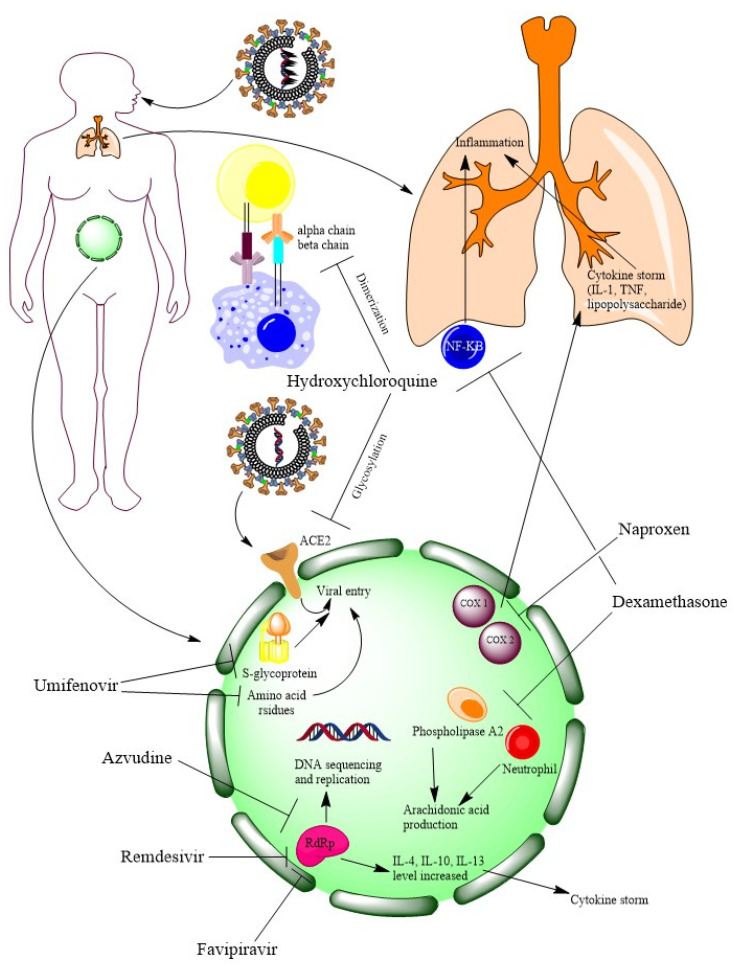
At a glance MOAs of repurposed drugs (figure is generated by ChemDraw Professional 15.1).

**Table 1 vaccines-11-00332-t001:** Some clinical trials on new molecules and materials for the treatment of COVID-19 (source: clinicaltrials.gov; accessed on 24. January 2023).

Clinicaltrials.gov Identifier	Nature of Clinical Trial	Selection Criteria of Volunteer	No. of Volunteers	Sponsor(s)	Drug Product(s)/Treatment	Phase	Final Outcome
NCT04392219	Randomized, double blind, and placebo controlled	Adults and older adults	130	Ridgeback Biotherapeutics at Covance Leeds Clinical Research Unit, UK	Molnupiravir	Phase 1	Molnupiravir was more tolerable than placebo, and adverse events were also lesser than placebo
NCT04405570	Randomized, double blind, and placebo controlled	Adults and older adults	204	Ridgeback and Merck	Molnupiravir	Phase 2	400 and 800 mg dose bid was effective with no side effects
NCT04575597	Randomized, double-blind, and placebo-controlled trial	Adults and older adults	1850	Merck	Molnupiravir	Phase 2/3	Effective in reducing death, 6.8% (95% CI) for the drug and 9.7% (95% CI) for placebo
NCT04960202	Randomized and quadruple-blinded trial	Adults and older adults	2246	Pfizer	Paxlovid	Phase 2/3	Less death risk and adverse effects; therapeutic efficacy maintained with percentage point difference of −5.81
(NCT05011513	Randomized, quadruple blinded, and parallel sequenced	Adults and older adults	1140		Paxlovid		Terminated due to lower number of deaths and hospitalized standard-risk patients
NCT04421027	Randomized, double blinded, and parallel assigned	Adults and older adults	1525	Eli Lilly	Baricitinib	Phase 3	Mortality was reduced by 38.2%
NCT04333368	Randomized, triple blinded, and parallel assigned	Adults and older adults	47	Assistance Publique–Hôpitaux de Paris	Wharton’s jelly mesenchymal stem cells	Phase 1/2	Adverse effects were reported in 28.6% and 25% of patients from the intervention and placebo group, respectively
NCT04321421	Open-label study	Adults and older adults, having moderate to severe ARDS	49	Foundation IRCCS San Matteo Hospital	Convalescent plasma		Result showed uncertainty about beneficial effect of CP
NCT04315298	Randomized, double blind, and placebo controlled	Adult or older adult patients having confirmed COVID-19 infection	1912	Regeneron Pharmaceuticals in collaboration with Sanofi	Sarilumab	Phase 2/3	The percentage of improvement was 43.2% and 35.5% in the intervention and placebo group, respectively
NCT04327388	Randomized, quadruple blinded, parallel mode	Adults or older adults	420	Sanofi in collaboration with Regeneron Pharmaceuticals	Sarilumab	Phase 3	Adverse effects were reported by 70%, 65%, and 65% of patients with 400 and 200 mg sarilumab and placebo, respectively
NCT04320615	Randomized, double blinded, parallel mode, and multicenter	Adult or older adult patients	452	Hoffmann-La Roche	Tocilizumab	Phase 3	No beneficial outcomes were obtained
NCT04275414	Open label	Adult or older adults having confirmed COVID-19 infection	27	Qilu Hospital of Shandong University	Bevacizumab	Phase 2	Body temperature was normalized in 93% of patients and improvement in PaO_2_/FiO_2_ ratios

**Table 2 vaccines-11-00332-t002:** Clinical trials of the some repurposed drugs for the treatment of COVID-19 (collected from clinicaltrials.gov; accessed on 24 January 2023).

Clinicaltrials.gov Identifier	Nature of Clinical Trial	Selection Criteria of Volunteer	No. of Volunteers	Sponsor(s)	Drug Product(s)/Treatment	Phase	Final Outcome
NCT04327401	Randomized, open label, parallel mode	Adult or older adults	290	Hospital Israelita Albert Einstein, Hospital do Coracao, Brazilian Research in Intensive Care Network and Ache Laboratorios Farmaceuticos S.A.	Dexamethasone	Phase 3	Terminated by the data monitoring committee for the unaccepted recovery results
NCT04325633	Open label, randomized, parallel model	Adult or older adult patients	584	Assistance Publique—Hôpitaux de Paris	Naproxen	Phase 3	Terminated due to insufficient recruitment of participants
NCT04292899	Randomized, open label, parallel	Hospitalized child, adult, or older adults	6000	Gilead Sciences	Remdesivir	Phase 3	After 14 days. 74.4% and 59% of patients survived from the intervention and noninterventional group.
NCT04280705	Randomized, double blinded, and parallel mode	Hospitalized adult or older adult patients	572	National Institute of Allergy and Infectious Diseases (NIAID)	Remdesivir	Phase 3	Adverse events were seen in 24.6% and 31.6% of patients in the interventional and placebo groups, respectively
NCT04332991	Randomized, quadruple blinded, parallel	Adult or older adult patients	510	Massachusetts General Hospital in collaboration with National Heart, Lung, and Blood Institute (NHLBI)	Hydroxychloroquine	Phase 3	No significant differences between the interventional group and the placebo group was found
NCT04346628	Randomized, open label, parallel mode	Adult or older adult patients	120	Stanford University	Favipiravir	Phase 2	No significant differences (95% CI) were observed between groups in mortality and symptom resolution
NCT04349241	Randomized, open label, parallel	Adult or older adult patients	100	Faculty of Medicine, Ain Shams University Research Institute- Clinical Research Center	Favipiravir	Phase 3	Two consecutive SARS-CoV-2-negative reports were observed after the treatment
NCT04252885	Randomized open label, parallel	Adult or older adult patients	125	Guangzhou Eighth People’s Hospital	Umifenovir	Phase 4	Arbidol monotherapy could be beneficial for mild, moderate, and severe COVID-19 patients

**Table 3 vaccines-11-00332-t003:** Approved repurposed drugs with their approved brand names.

Drug	Formulation	Approved by and Type of Approval	Date of Approval	Marketed Name	References
New Drugs					
Baricitinib	Oral tablets	USFDA (EUA)	October 2022	Olumiant^®^	[43]
Casirivimab and Imdevimab	Injectable solution	FDA (EUA), EMA (full approval)	November 2020 (USFDA), February 2021 (EMA for REGN-COV2), October 2021 (EMA for Ronapreve)	REGEN-COV^®^ (USFDA), REGN-COV2 (EMA), Ronapreve (EMA)	[167,168,169]
Molnupiravir	Oral capsules	USFDA (EUA), MHRA, EMA (EUA)	December 2021 (USFDA), November 2021 (MHRA, EMA)	Lagevrio™	[26,27,28,29]
Nirmatrelvir and Ritonavir	Oral tablets	USFDA (EUA), EMA (EUA)	December 2021 (USFDA, EMA)	Paxlovid™	[33,37]
Tocilizumab	Injectable solution	USFDA (full approval)	December 2022	Actemra^®^	[85]
Repurposed Drugs					
Favipiravir	Tablets	DCGI (EUA)	July 2020	Fabiflu^®^	[137]
Hydroxychloroquine	Oral formulation	USFDA (EUA), EMA (EUA)	March 2020April 2020		[124,125,126,127]
Remdesivir	Injection, solution	USFDA (EUA), EMA (EUA)	October 2020December 2021	Veklury^®^	[113,114,115]
Tixagevimab and Cilgavimab	Injection, solution	USFDA (EUA)	December 2021	Evusheld™	[170]

## Data Availability

Not applicable.

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
