# Peer review of "A Clinical Insight on New Discovered Molecules and Repurposed Drugs for the Treatment of COVID-19"

_vaccines, 2023, doi:10.3390/vaccines11020332_

Round 1

Reviewer 1 Report

Comments to the Authors:

The article "New Discovered Molecules and Repurposed Drugs for the treatment of Covid-19" reviews the information on new molecules and existing drugs which have been used and/or undergone clinical trials as therapeutics for COVID-19. The authors did not provide clear review objectives which might lead to lengthy and unfocused discussions. This review is too long and wordy with too many details presented. Then, the authors should simplify and make the paragraphs concise with only relevant information. The authors should put their analysis in this review by grouping and comparing data and not just putting the information directly from the references. Moreover, the authors should provide significant discussion and conclusion, giving new insights and knowledge than the previous similar reviews. Next, the authors may create graphs to add novelty and value to this article.

1.     The article's title is clear. However, the authors may revise the title to emphasize the novelty offered by this review article.

2.     The authors have provided the objective of the review in the abstract. However, they did not explain this clearly in the introduction.

3.      Introduction,

The introduction is too wordy and contains unnecessary information. Furthermore, It is not cohesive and flowing with the idea between paragraphs is not interconnected. Moreover, the paragraphs are too long, with too many ideas within the paragraph. For example, paragraph 1 contains three main ideas. The introduction also has a repetition of information which many information appear later in the body of the review.

3.1    Some of the sentences need to rewrite as they are confusing and inaccurate (lines 38-39, 43-45, anti-covid 19 vaccines?)

3.2    The authors should revise their referencing as many sentences in the paragraphs seem to miss the citation.

3.3    The authors may state the novelty and the valuable aspects offered by this review compared to previously available reviews article.

4.   The authors should revise the sections heading (line 116 and line 480) as the term is used differently with the title and the introduction.

5.   Section 2 New Drugs:

5.1              The authors should define what the "new drugs/molecules" mean.

5.2              This section is too wordy, with several long paragraphs. The authors should revise it by making it concise and clear. The authors should have a pattern to present each drug's aspect. Therefore the reader will quickly understand the difference between drugs. There is no need to put the details of the clinical trials as they can be accommodated in Table 1.

5.3              The authors should analyze the information from the articles they reviewed as it will add significant value to this article.

5.4              The authors should also revise the referencing in this section.

6.     Section 3 Pre-existing Drugs

6.1              Several paragraphs of this section are too long, with too many ideas.

6.2              There is no need to put the details of the clinical trials as they can be accommodated in Table 2.

6.3              The authors should analyze the information from the articles they reviewed as it will add significant value to this article

7.     The authors should describe the limitation of this review

8.      Conclusion

The conclusion should be based on the analysis of the review. The authors should revise the conclusion as some sentences contradict (lines 906-907 and 919-921). Furthermore, the authors may put recommendations and ways forward for the future.

9.     Referencing

The authors should use the proper referencing methods and citations.

10.  The authors may create graphs instead of using previously published graphs, which may present the differences of the drugs discussed.

11.  Table 1 is too huge. Instead, the authors may divide the content into two tables, with one table as the supplementary table.

Author Response

Dear Reviewer 1

The authors are thankful to reviewers and the editorial office for spending their valuable time in providing suggestions in improving the quality of this work. We have tried to address all of these suggestions in this revision. Our approaches to handle the revision on specific concerns raised by reviewers are given as under:

Answers to comments of Reviewer 1:

  1. The authors did not provide clear review objectives which might lead to lengthy and unfocused discussions.

Ans: Clear review objectives are provided both in abstract and introduction. These are also discussed in introduction and are underlined in the manuscript. The entire review is now focused to only these objectives. The content is drastically reduced and only clinical trials with significant results have been focussed.

  1. This review is too long and wordy with too many details presented.

Ans: The entire review is now focused to set objectives. The content is drastically reduced and only clinical trials with significant results have been incorporated.

Earlier the manuscript had 60 pages which have been reduced to 40 pages in this revision process.

  1. Then, the authors should simplify and make the paragraphs concise with only relevant information.

Ans: The entire manuscript is revised to simplify and making the content concise. A track change version and a clean version is submitted for further review.

  1. The authors should put their analysis in this review by grouping and comparing data and not just putting the information directly from the references.

Ans: Both new discovered drugs and repurposed drugs are analysed comparatively in two separate sections viz. Section 3: Comparative analysis of new discovered molecules and Section 5: Comparative analysis of new purposed drugs.

  1. Moreover, the authors should provide significant discussion and conclusion, giving new insights and knowledge than the previous similar reviews.

Ans: A comparative discussion on drugs is added in new sections, viz.Section 3: Comparative analysis of new discovered molecules and Section 5: Comparative analysis of new purposed drugs.

Conclusion is also revised in the light of suggestions.

  1. Next, the authors may create graphs to add novelty and value to this article.

Ans:The following new figures and tables have been added to add novelty and value to this article.

Figure 1. Comparison of survival rate of COVID-19 patients in clinical trials conducted with new discovered molecules.

Figure 2. Comparison of adverse events observed in COVID-19 patients in clinical trials conducted with new discovered molecules.

Figure 3. At a glance MOAs of new discovered molecules for COVID-19.

Figure 4. Comparison of survival rates of COVID-19 patients in clinical trials conducted with repurposed drugs.

Figure 5. Comparison of adverse events observed in COVID-19 patients in clinical trials conducted with repurposed drugs.

Figure 6. At a glance MOAs of repurposed drugs.

Table 3.Approved repurposed drugs with their approved brand names.

  1. The article's title is clear. However, the authors may revise the title to emphasize the novelty offered by this review article.

Ans:The article title is now revised to “A Clinical Insight on New Discovered Molecules and Repurposed Drugs for the Treatment of COVID-19” to emphasize the novelty of the article.

  1. The authors have provided the objective of the review in the abstract. However, they did not explain this clearly in the introduction.

Ans: Objectives are now provided clearly in the last paragraph of introduction. This content is now underlined in the revised manuscript.

  1. Introduction,
    • The introduction is too wordy and contains unnecessary information.

Ans: Introduction is now concise and unnecessary information is deleted.

  • Furthermore, it is not cohesive and flowing with the idea between paragraphs is not interconnected.

Ans: Paragraphs are rephrased, sentences are reconstructed and made interconnected.

  • Moreover, the paragraphs are too long, with too many ideas within the paragraph. For example, paragraph 1 contains three main ideas.

Ans:Paragraphs are now concise. The first paragraph now introduces the COVID-19 disease and treatment approaches.

  • The introduction also has a repetition of information which many information appear later in the body of the review.

Ans: Repeated information are deleted.

  • Some of the sentences need to rewrite as they are confusing and inaccurate (lines 38-39, 43-45, anti-covid 19 vaccines?)

Ans:The confusing and inaccurate sentences are either deleted or rephrased.

  • The authors should revise their referencing as many sentences in the paragraphs seem to miss the citation.

Ans: Referencing have been revised in the paragraphs.

  • The authors may state the novelty and the valuable aspects offered by this review compared to previously available reviews article.

Ans: The novelty and the valuable aspects, offered by this review compared to previously available reviews article, is stated in the introduction.

  1. The authors should revise the sections heading (line 116 and line 480) as the term is used differently with the title and the introduction.

Ans: The section headings (line 116 and line 480) are revised as per title and introduction.  These lines have been rephrased.

  1. Section 2 New Drugs:
    • The authors should define what the "new drugs/molecules" mean.

Ans: "new drugs/molecules" are defined in the first three lines of Section 2. These lines have been underlined in the manuscript.

  • This section is too wordy, with several long paragraphs. The authors should revise it by making it concise and clear.

Ans: This section is now concise.

  • The authors should have a pattern to present each drug's aspect. Therefore the reader will quickly understand the difference between drugs.

Ans: Each drug is now described as per objective of the review and separate sub-sections have been created.

  • There is no need to put the details of the clinical trials as they can be accommodated in Table 1.

Ans: Important details of clinical trials for the drugs are discussed in text and at a glance they are enlisted in Table 1. Table 1 is shortened now.

  • The authors should analyze the information from the articles they reviewed as it will add significant value to this article.

Ans: To add significant value entire manuscript is revised and two separate sections (Section 3 and Section 5) have been added to critically analyze the information and compare various drugs.

  • The authors should also revise the referencing in this section.

Ans: All references have been checked and revised now.

  1. Section 3 Pre-existing Drugs
    • Several paragraphs of this section are too long, with too many ideas.

Ans: The content is focused and has been reduced.

  • There is no need to put the details of the clinical trials as they can be accommodated in Table 2.

Ans: Important details of clinical trials for the drugs are discussed in text and at a glance they are enlisted in table 2.

  • The authors should analyze the information from the articles they reviewed as it will add significant value to this article.

Ans: Two separate sections (Section 3 and Section 5) have been added to critically analyze the information and compare various drugs.

  1. The authors should describe the limitation of this review.

   Ans: Limitations of this review are added in conclusion.

  1. Conclusion
    • The conclusion should be based on the analysis of the review.

Ans: Conclusion is rephrased and revised.

  • The authors should revise the conclusion as some sentences contradict (lines 906-907 and 919-921).

Ans: Contradictory lines are deleted and conclusion is revised.

  • Furthermore, the authors may put recommendations and ways forward for the future.

Ans: Future directions are now provided in conclusion.

  1. Referencing
    • The authors should use the proper referencing methods and citations.

Ans: Referencing and citations are corrected.

  1. The authors may create graphs instead of using previously published graphs, which may present the differences of the drugs discussed.

Ans: The following new figures and tables have been added to add novelty and value to this article.

Figure 1. Comparison of survival rate of COVID-19 patients in clinical trials conducted with new discovered molecules.

Figure 2. Comparison of adverse events observed in COVID-19 patients in clinical trials conducted with new discovered molecules.

Figure 3. At a glance MOAs of new discovered molecules for COVID-19.

Figure 4. Comparison of survival rates of COVID-19 patients in clinical trials conducted with repurposed drugs.

Figure 5. Comparison of adverse events observed in COVID-19 patients in clinical trials conducted with repurposed drugs.

Figure 6. At a glance MOAs of repurposed drugs.

Table 3.Approved repurposed drugs with their approved brand names.

  1. Table 1 is too huge. Instead, the authors may divide the content into two tables, with one table as the supplementary table.

Ans: Table 1 is modified and shortened.

We look forward to hearing from you soon on outcome of this revision.

With thanks and kind regards,

Vikas Anand Saharan and Hitesh Kulhari

Reviewer 2 Report

The manuscript by Banerjee et al described a review on new discovered molecules and repurposed drugs for treatment of COVID-19. The manuscript is well-orgranized with a few comments:

1.  Azvudine should be included in the category of repurposed drugs, which has been approved by SFDA.

2. In line 19, COVID-19 was first detected in December 2019, not in November 2019.

3. The abbreviations SC-2 and C -19 are not widely accepted. SARS-CoV-2 and COVID-19 are recommened, respectively.

4. The authors should avoid using subjective adjective, for example gruesome in line 43. 

Author Response

The authors are thankful to reviewers and the editorial office for spending their valuable time in providing suggestions in improving the quality of this work. We have tried to address all of these suggestions in this revision. Our approaches to handle the revision on specific concerns raised by reviewers are given as under:

ANSWERS TO COMMENTS OF REVIEWER 2:

  1. Azvudine should be included in the category of repurposed drugs, which has been approved by SFDA.

Ans: Azvudine is included in the category of repurposed drugs.

  1. In line 19, COVID-19 was first detected in December 2019, not in November 2019.

Ans: This information is corrected.

  1. The abbreviations SC-2 and C -19 are not widely accepted. SARS-CoV-2 and COVID-19 are recommened, respectively.

Ans: Abbreviations SC-2 and C-19 are changed to SARS-CoV-2 and COVID-19, respectively.

  1. The authors should avoid using subjective adjective, for example gruesome in line 43. 

Ans: Subjective objective, used in text, is deleted.

We look forward to hearing from you soon on outcome of this revision.

With thanks and kind regards,

Vikas Anand Saharan and Hitesh Kulhari

Round 2

Reviewer 1 Report

Overall, most questions and comments have been addressed by the authors. However, there is some revision required, such as:

1.     Line 42-44

“A grave challenge with COVID-19 is the continuous mutation of SARS-COV-2 virus leading to several  variants (α, β, γ, δ, etc.) and their strains (B.1.1.7, P.1, B.1.351, B.1.617.2 etc.) spread all over the world [2,3]”.

This sentence has inaccurate information as “B.1.1.7, P.1, B.1.351, B.1.617.2” are not the strains of variants (α, β, γ, δ). Variant α is variant B.1.1.7. See https://www.who.int/activities/tracking-SARS-CoV-2-variants

2.     Line 402

Please, fix the typo.

Author Response

Dear Reviewer (Round 2),

The authors are thankful to reviewer for spending their valuable time in providing suggestions in improving the quality of this work. We have tried to address all of these suggestions in this revision. Our approaches to handle the revision on specific concerns raised by reviewers are given as under:

 Answers to comments of Reviewer 1 (Round 2):

  1.    Line 42-44

“A grave challenge with COVID-19 is the continuous mutation of SARS-COV-2 virus leading to several  variants (α, β, γ, δ, etc.) and their strains (B.1.1.7, P.1, B.1.351, B.1.617.2 etc.) spread all over the world [2,3]”.

This sentence has inaccurate information as “B.1.1.7, P.1, B.1.351, B.1.617.2” are not the strains of variants (α, β, γ, δ). Variant α is variant B.1.1.7. See https://www.who.int/activities/tracking-SARS-CoV-2-variants

Ans: We sincerely thank the reviewers for pointing out this mistake and providing the most suitable reference in the comment. We have studied the details of different SARS-CoV-2 variants (including different nomenclatures like GISAID, Nextstrain, Pango) from the WHO official website (as mentioned in the comment). The expert group later recommended to use some Greek alphabets (like α, β, γ, δ) to recognize the pango lineages for the non-scientific audiences. For this reason, α variant is variant B.1.1.7.We are sorry for this mistake in our manuscript (line 42-44). The sentence in line 42-44 is corrected and the pango lineages are only retained in it.

  1.    Line 402

Please, fix the typo.

Ans: Typographical mistake in the line 402 is corrected.

We look forward to hearing from you soon on outcome of this revision.

With thanks and kind regards,

Vikas Anand Saharan and Hitesh Kulhari